# Quantitative and Qualitative Analysis of Multicomponent Gas Using Sensor Array

**DOI:** 10.3390/s19183917

**Published:** 2019-09-11

**Authors:** Shurui Fan, Zirui Li, Kewen Xia, Dongxia Hao

**Affiliations:** Tianjin Key Laboratory of Electronic Materials Devices, School of Electronics and Information Engineering, Hebei University of Technology, Tianjin 300401, China; 201821902031@stu.hebut.edu.cn (Z.L.); kwxia@hebut.edu.cn (K.X.); 201831903026@stu.hebut.edu.cn (D.H.)

**Keywords:** gas sensor array, cross-sensitivity, PCA, random forest, particle swarm optimization

## Abstract

The gas sensor array has long been a major tool for measuring gas due to its high sensitivity, quick response, and low power consumption. This goal, however, faces a difficult challenge because of the cross-sensitivity of the gas sensor. This paper presents a novel gas mixture analysis method for gas sensor array applications. The features extracted from the raw data utilizing principal component analysis (PCA) were used to complete random forest (RF) modeling, which enabled qualitative identification. Support vector regression (SVR), optimized by the particle swarm optimization (PSO) algorithm, was used to select hyperparameters *C* and *γ* to establish the optimal regression model for the purpose of quantitative analysis. Utilizing the dataset, we evaluated the effectiveness of our approach. Compared with logistic regression (LR) and support vector machine (SVM), the average recognition rate of PCA combined with RF was the highest (97%). The fitting effect of SVR optimized by PSO for gas concentration was better than that of SVR and solved the problem of hyperparameters selection.

## 1. Introduction

Gas is everywhere in our lives. The gas exhaled by humans contains a labeled gas that can indicate certain diseases. For example, a large amount of acetone appears in the exhalation of a diabetic patient [1], a large amount of ammonia appears in the exhalation of a uremic patient [2], and surfaces produce fungi and volatilize organic compounds after food deteriorates [3,4]. The generation of gas is closely related to changes occurring in the substances around it. Since it can be used as a basis for analyzing such changes, gas detection is particularly important.

Gas sensor arrays associated with machine learning algorithms are widely used in different fields, such as the use of an electronic nose to judge the quality of food [5], predict food additives in juice [6], evaluate paraffin samples [7], classify different essential oils [8], monitor air quality using drones in real time [9,10,11,12,13,14], analyze the spatial distribution of air pollutants [15], and predict future air quality [16,17]. In addition, they can be used to determine leak sources based on the gas concentration distribution [18]. However, the gas sensor element has cross-sensitivity, which makes it is impossible to use a single gas sensor to effectively detect the composition of a gas mixture.

In light of this problem, a wide variety of machine learning algorithms have been used for gas identification or gas quantification, including kernel principal component analysis (KPCA) [19], linear discriminant analysis (LDA) [8], logistic regression (LR) [20], support vector machines for classification and regression (SVM [7] and SVR [16,21]), artificial neural networks (ANN) [14], and reservoir computing [22]. It is also a good method to select the appropriate parameters from the sensor response signals for the identification and concentration estimation of mixed gas [4,23]. A summary of gas mixture analysis methods is shown in Table 1. There are three works on qualitative identification (QAL-ID), which have been applied to gas mixtures, volatile gas of paraffin, and essential oils. There are also reports on quantitative analysis (QTY-ANLS), which were applied to gas mixtures, emissions of LNG (liquefied natural gas) bus or food additives in fruit juice, and air pollutants.

Although the above strategies can, to a certain extent, be effectively used for mixture detection and prediction of concentration, they still pose problems. KPCA requires choosing the appropriate kernel function and parameter *ξ*, which reduces the training efficiency. The classification model based on ANN requires a large number of training samples to achieve good training results, and is prone to overfitting and local optimum. In addition, the structure of a neural network is generally determined by an empirical method, which leads to a certain degree of gas identification accuracy decline. In many previous works, SVR has been shown to outperform other competing methods in regression tasks for gas quantification [24,25]. However, the hyperparameters of this algorithm are determined using the grid search method [6], which traverses the subspace of the specified value parameter to select the optimal value. Since the value space of a hyperparameter is not restricted, in many cases, any real value can be taken, and the choice of the subspace is not simple.

To avoid such problems, the main objective of this study was to propose a gas mixture analysis method to be applied to a gas sensor array. This proposal must include qualitative identification and quantitative analysis for gas mixtures, which would make it possible to use a gas sensor array to effectively detect the composition of a gas mixture.

## 2. Gas Mixture Analysis Method

As shown in Figure 1, the mixed gas analysis method proposed in this paper is mainly divided into two parts: qualitative identification and quantitative analysis. We used PCA combined with random forest (RF) as a tool for qualitative identification, and the quantitative analysis adopted SVR optimized by the particle swarm optimization (PSO) algorithm (PSO + SVR).

For the qualitative identification, we chose 1/10 of a dataset to determine the number of principal components. After that, 9/40 of the dataset were extracted features utilizing PCA, which was used to build the random forest model. Then, the feature set extracted from another 9/40 was used for testing the generated model, through which we obtained the identification results. For the quantitative analysis, 9/40 of the dataset was used for training the optimization model (PSO + SVR), after which the combination of *C* and *γ* are obtained. By applying the combination of *C* and *γ* to SVR, we obtained the regression model. Finally, the quantitative analysis results for the last 9/40 of the dataset were obtained.

## 3. Qualitative Identification Method for Gas Mixture

### 3.1. Principal Component Analysis

Feature extraction is an important topic and the basis of pattern recognition and machine learning [30]. Principal component analysis is a method of feature extraction. The basic idea of it is to transform the original features into a group of new features in order of importance, from the largest to the smallest, through a set of orthogonal vectors [29]. These new features are linear combinations of the original features and they are unrelated to each other. We have provided a working process of principal component analysis.

Consider the original sample X=[x1,x2,…,xM]∈ℝM×N, where *N* is the number of variables, *M* is the number of samples, and xi∈ℝN(iϵM) represents the *i*th *N*-dimensional sample.

Firstly, the data of each dimension are decentralized. That is, the characteristics of each dimension are subtracted from their average values, as shown in Equation (1). $x_{i}^{j}(i\in M,j\in N)$ is the *i*th sample of the *j*th variable, and $x_{i}^{j*}$ is the decentralized value of $x_{i}^{j}$. Secondly, the covariance matrix of $X^{*}$ is calculated using Equation (2), and the eigenvalues and eigenvectors of it are obtained by eigenvalue decomposition. Then, the eigenvalues are sorted from largest to smallest as $\lambda_{1},\;\lambda_{2},\;\ldots ,\;\lambda_{N}$, and the corresponding eigenvectors are $\alpha_{1},\alpha_{2},\ldots ,\alpha_{N}$. Finally, the reduced number p is determined by the cumulative contribution rate of the eigenvalue for variance $r_{CCR}$ (Equation (3)), which utilizes $r_{CCR}\geq 99\%$:(1)$$x_{i}^{j*}=x_{i}^{j}-\frac{x_{1}^{j}+x_{2}^{j}+\ldots +x_{M}^{j}}{M};$$
(2)$$C=\frac{1}{M}X^{*}X^{*T};$$
(3)$$r_{CCR}=\frac{\sum_{i=1}^{p}\lambda_{i}}{\sum_{j=1}^{N}\lambda_{j}}\times 100.$$

### 3.2. Random Forest

The random forest method comes from the decision tree and bagging methods. The decision tree learns a model from the given training dataset to classify new samples. The algorithm needs two sets of data: the training data used to construct the decision mechanism and the test data used to verify the constructed decision tree. The process of the decision tree learning algorithm (Algorithm 1) is presented below.


**Algorithm 1. Decision Tree**
Input: Training set $D=\left\{\left(x_{1},y_{1}\right),\left(x_{2},y_{2}\right),\ldots ,\left(x_{m},y_{m}\right)\right\};$   Attribute set $A=\left\{a_{1},a_{2},\ldots ,a_{d}\right\}$Process: Function Tree Generate (D, A)1.Generate the node
2.If all samples in D belong to the same category C, then

3. Mark node as leaf node of class C

4.End if

5.If A = ∅ or the samples in D have the same value in A, then

6. Mark node as leaf node, its category is marked as the class with the largest number of samples in D; return

7.End if

8.Choose the optimal partition properties $a_{*}$

9.For every value $a_{*}^{v}$ of $a_{*}$, do

10. Generate a branch for node; let $D_{v}$ represent the sample subset of D evaluated at $a_{*}$ to the $a_{*}^{v}$

11. If $D_{v}$ is empty, then

12.  Mark branch node as leaf node, its category is marked as the class with the largest number of samples in D; return

13. Else

14.  Take TreeGenerate($D_{v}$,A\{$a_{*}$}) as branch node

15. End if

16.End for
Output: A decision tree with root node

On the basis of the bagging integration decision tree, the random forest further introduces random attribute selection in the training process of the decision tree.

## 4. Quantitative Analysis Method for Gas mixture

### 4.1. Support Vector Regression

Support vector regression is an important application branch of support vector machine. The basic idea is to find a regression plane to which all the data of a set are closest.

Consider training samples D={(x1,y1),(x2,y2),…(xm,ym)},yi∈ℝ (*m* is the number of the samples), which aims to learn a regression model shaped like Equation (4), so that *f*(*x*) is as close as possible to *y* (the absolute value), and *ω* and *b* are the model parameters to be determined:(4)f(x)=ωTx+b.

Suppose we can tolerate maximum deviation *ε* between *f*(*x*) and *y*, namely, only when the difference between *f*(*x*) and the absolute value *y* is larger than *ε* is the loss calculated. This is equivalent to making *f*(*x*) the center, built with a width of 2*ε* intervals, and if the training samples are in this interval, the prediction will be right. We can obtain a loss function *g*(*n*) with Equation (5) (*N* is the number of samples, $y_{n}$ is the true value, and $t_{n}$ is the predicted value):(5)$$g\left(n\right)=\frac{1}{2}\sum_{n=1}^{N}{\left\{y_{n}-t_{n}\right\}}^{2}+\frac{1}{2}{\left\|\omega \right\|}^{2}.$$

The optimization problem can be re-expressed by introducing relaxation variables *ε*. For each data point $x_{n}$, the condition which makes the prediction point locate in the interval band is Equation (6), and the points above and below the interval satisfy Equation (7), where $y\left(x_{n}\right)$ is the true value, and $\zeta_{n}$ and $\hat{\zeta_{n}}$ are the positive and negative values of $t_{n}$ beyond the interval 2ϵ:(6)$$y_{n}-\epsilon \leq t_{n}\leq y_{n}+\epsilon ,$$
(7)$$t_{n}\leq y\left(x_{n}\right)+\epsilon +\zeta_{n}\;and\;t_{n}\geq y\left(x_{n}\right)-\epsilon -\hat{\zeta_{n}}.$$

The optimization problem of support vector regression can be written as Equation (8):$$\underset{\omega ,b,\zeta_{n},\hat{\zeta_{n}}}{\min}C\overset{N}{\underset{n=1}{\sum }}\left(\zeta_{n}+\hat{\zeta_{n}}\right)+\frac{1}{2}{\left\|\omega \right\|}^{2}$$
(8)$$s.t.\;\begin{array}{c} t_{n}\leq y\left(x_{n}\right)+\varepsilon +\zeta_{n}\\ t_{n}\geq y\left(x_{n}\right)-\varepsilon -\hat{\zeta_{n}}\\ \zeta_{n}\geq 0,\hat{\zeta_{n}}\geq 0,n=1,\ldots ,N. \end{array}$$

### 4.2. PSO

Particle swarm optimization seeks the optimal solution through cooperation and information sharing among individuals in the group. It simulates the swarm behavior of insects, herds, birds, and fish, which search for food in a cooperative way, with each member of the group constantly changing its search patterns by learning from its own experience and that of other members. The whole process of the algorithm is as follows: *Step* *1.*Initialize a group of particles with the group size *n*, set their original velocity and location, and set the maximum number of iterations at the same time;*Step* *2.*Define the fitness function to evaluate the fitness of each particle;*Step* *3.*Find the optimal solution for each particle (individual extremum), from which a global value is found, which is called the global optimal solution;*Step* *4.*Update the velocity and position of the particle by Equations (9) and (10), where *V_id_* and *X_id_* are the *d* dimensional velocity and position of particle *i*, *P_id_* and *P_gd_* are the *d* dimensional optimal position searched by particle *i* and the global optimal position of the whole group, ω is the inertia factor, *C*_1_ and *C*_2_ are the learning factor, and random(0, 1) is a random number between (0, 1):(9)$$V_{id}=\omega V_{id}+C_{1}random\left(0,1\right)\left(P_{id}-X_{id}\right)+C_{2}random\left(0,1\right)\left(P_{gd}-X_{id}\right),$$
(10)$$X_{id}=X_{id}+V_{id}.$$*Step* *5.*The algorithm will be terminated when the number of iterations reaches the setting; otherwise, it will return to step 2 to continue execution.

### 4.3. SVR Optimized by the PSO Algorithm

The performance of SVR depends on the appropriate choice of hyperparameters *C* and *γ*. The penalty coefficient *C* reflects the degree of the algorithm’s penalty on the sample data exceeding the *ε* pipelines, and its value affects the complexity and stability of the model. If *C* is too small, the penalty for the sample data exceeding *ε* pipelines is small and the training error becomes larger. If *C* is too large, the learning accuracy will be improved correspondingly, but the generalization ability of the model will be worse. *γ* reflects the degree of correlation between the support vectors. If it is very small, the connection between the support vectors is relatively loose, learning machines are relatively complex, and promotion ability cannot be guaranteed; on the other hand, if it is too large, the influence between support vectors will be too strong, and the regression model will have difficulty achieving sufficient accuracy.

Particle-swarm-optimized SVR was used here to select the optimal combination of *C* and *γ*, which can solve the problem of hyperparameter selection and improve the prediction accuracy. The algorithm flow of particle-swarm-optimized SVR is shown in Figure 2.

The algorithm steps were as follows: *Step* *1.*Import the original data, divide it into training data and test data, and normalize these;*Step* *2.*Initialize the parameters of PSO, including population *n*, particle velocity *v*, and position *x*, and iteration number;*Step* *3.*Calculate the fitness value of the particle. The current fitness value of the particle is compared with the fitness value of the historically optimal position. If it is better, it will be regarded as the current optimal position. Compared with the global optimal position fitness value of each particle, if it is better, it will be considered the current global optimal position;*Step* *4.*Update the velocity and position of the particle by Equations (9) and (10);*Step* *5.*Determine whether termination conditions are met. If they are satisfied, the optimal *C* value and γ value are output and assigned to SVR. Otherwise, return to step 3;*Step* *6.*Test the optimal model of SVR and obtain the prediction results.

## 5. Experiments and Results

### 5.1. Dataset

The dataset used was based on the UCI (University of California Irvine) dataset [25], which consists of the responses of methane, ethylene, air, and their mixtures in arrays of 16 sensors (TGS2600, TGS2602, TGS2610, and TGS2620; four units of each type) with a continuous measurement time of 10,486 s. The gas-sensing material of this type of gas sensor is a metal oxide which is adsorbed on the surface of the metal oxide when it is heated to a certain high temperature in the air. When a reducing gas occurs, the surface concentration of the negatively charged oxygen decreases, causing the resistance of the sensor to decrease. Some parameters of these four types of sensors are presented in Table 2.

In order to facilitate observation, the sensor responses and concentration values were normalized, as shown in Figure 3. The four channels from top to bottom were TGS2602, TGS2600, TGS2610, and TGS2620, as well as the concentration of two gases. As shown in Figure 3, TGS2602 responded significantly to changes in ethylene concentration, but the response curve was not very obvious. TGS2600 and TGS2620 responded to changes in methane and ethylene, TGS2610 responded significantly to changes in methane concentration, and the four sensors had slow responses to rapidly changing gases.

### 5.2. PCA Feature Extraction

The data matrix of PCA was 10,476 rows and 16 columns. The input matrix was artificially scaled so that the mean was 0 and the variance was 1. The covariance matrix of the normalized data was calculated to obtain a matrix of 16 rows and 16 columns. The 16 eigenvalues and contribution rates are shown in Table 3. When there were four principal components, the cumulative contribution rate reached 99.69% (more than 99%), which could almost represent all the information. From the fifth principal component, the cumulative contribution rate increased with a smaller step and gradually approached zero. It can be confirmed that the dataset decreased from the original 16 dimensions to four dimensions. In the literature [7], the volatile gas characteristics of paraffin samples were also analyzed using PCA. Using the first three principal components, it can be seen that the paraffin samples were clearly divided into four grades, and the eigenvalue contribution rate of these principal components was 93.34%, but the first five principal components were finally extracted to form a new feature dataset.

### 5.3. Qualitative Identification for Gas Mixture

The feature vector sets of training data were used to model the random forest, and then the feature vector sets of the test data were qualitatively identified by the model. In order to confirm the relevance of the random forest algorithm, we compared this algorithm with two other algorithms: LR and SVM. We chose these algorithms in their basic form. In our comparisons, we used the default parameters of each algorithm, as cited below: LR: penalty: ‘l2’, C: ‘1’, solver: ‘lbfgs’, multiclass: ‘multinomial’;SVM: ‘kernel: ‘linear’, decision_function_shape: ‘ovo’.

Figure 4 shows the confusion matrices for three classifiers, which were able to separate the four classes. In [31], they also used a confusion matrix to visually compare two classifiers. The sum of all values in the matrix is the total amount of data for classification. The values on the diagonal are the correctly identified data of each category, while the values off the diagonal are the misidentified data of each category. Comparing all of the data on the diagonal lines of the three figures (Figure 4a–c), we can see that the value on the diagonal of the RF is the largest, indicating that RF had the highest probability of correctly identifying each class compared with LR and SVM. The values on the diagonal of LR and SVM were similar, indicating that the classification effect of LR and SVM was similar. From Figure 4, we can see that the RF confusion matrix had the highest values on the diagonal and the lowest values off the diagonal. We calculated the average recognition rate *η* for each classifier using Equation (11) ($x_{ii}$ is the value on the diagonal, i∈1,2,3,4; *x* is the total classification data). We found that *η* of RF was the highest (97%), and the average recognition rate of LR and SVM was 85%, which is 12% less than RF:(11)$$\eta =\frac{x_{11}+x_{22}+x_{33}+x_{44}}{x}\times 100\%$$

### 5.4. Quantitative Analysis for Gas Mixture

Quantitative analysis for a gas mixture should be carried out after qualitative identification, in which the concentration estimation for a single gas and mixed gas by optimized SVR is carried out.

The number of particles and iterations were 81 and 10, respectively. The kernel function of SVR was selected as “rbf”, and 10-fold cross-validation was used in the training process. First, the training set was used for model training, and the best combination of *C* and *γ* was selected. Then, the test set was used for testing, and the concentration estimation results were obtained, with the determination coefficient *R*^2^ (Equation (12), in which ${\hat{y}}_{i}$ is the estimation value, $\bar{y}$ is the average of the actual concentration, and $y_{i}$ is the actual value) of the test samples as the evaluation criteria of the model to estimate ability. The value of *R*^2^ is between 0 and 1, and the closer it is to 1, the better the regression model. The selected values of *C* and *γ* in different categories are shown in Table 4. The prediction effects of the four classes based on SVR improved by PSO model are shown in Figure 5. It can be seen from the fitting curve in Figure 5 that the gas concentration fitting effects for the four categories were very good, except for the deviation of some sample points. The errors between the predicted and actual values fluctuated around 0, which indicates that the fitting effects were very good. In [7], they also used this way to compare three feature extraction methods.
(12)$$R^{2}=\frac{\sum {\left({\hat{y}}_{i}-\bar{y}\right)}^{2}}{\sum {\left(y_{i}-\bar{y}\right)}^{2}}.$$

In order to prove the regression effect of SVR optimized by the PSO algorithm, Figure 6 shows the comparison between our proposed methodology and SVR. It can be seen from Figure 6 that the approach based on SVR optimized by the PSO algorithm provided a smaller prediction error than SVR, which proves that the regression effect does improve through our method.

## 6. Conclusions

In this work, a novel qualitative and quantitative analysis strategy was proposed to provide accurate analysis of multicomponent gas mixtures. The proposed strategy combined PCA with random forest (PCA + RF) for identification. PCA can extract the principal components that contain most of the information and reduce the redundant factors. Random forest, as a classifier, was used to identify the gas mixture. The methodology also used SVR optimized by PSO as a tool to quantify the gas component of a mixture.

The experimental results show that the best identification performance was obtained by PCA + RF compared with LR and SVM. Its recognition rate was 97%, a gain of 12% compared with LR and SVM. SVR optimized by PSO had a better regression result for every gas component than SVR, and at the same time, it solved the problem of selecting the hyperparameters of SVR.

## Figures and Tables

**Figure 1 sensors-19-03917-f001:**
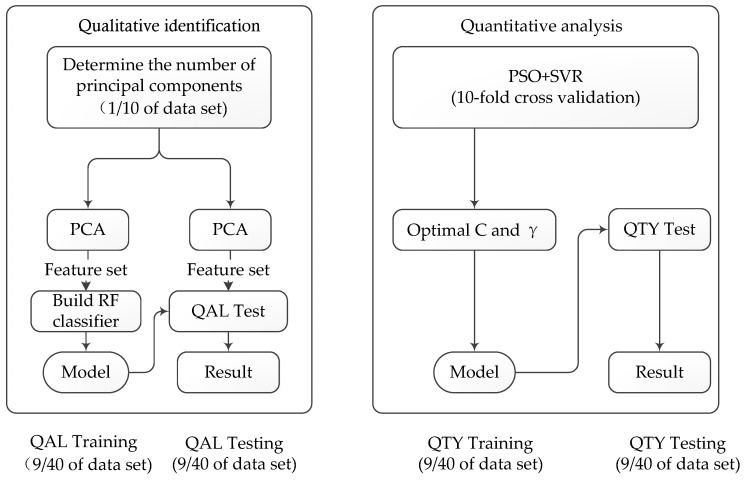
Gas mixture analysis method.

**Figure 2 sensors-19-03917-f002:**
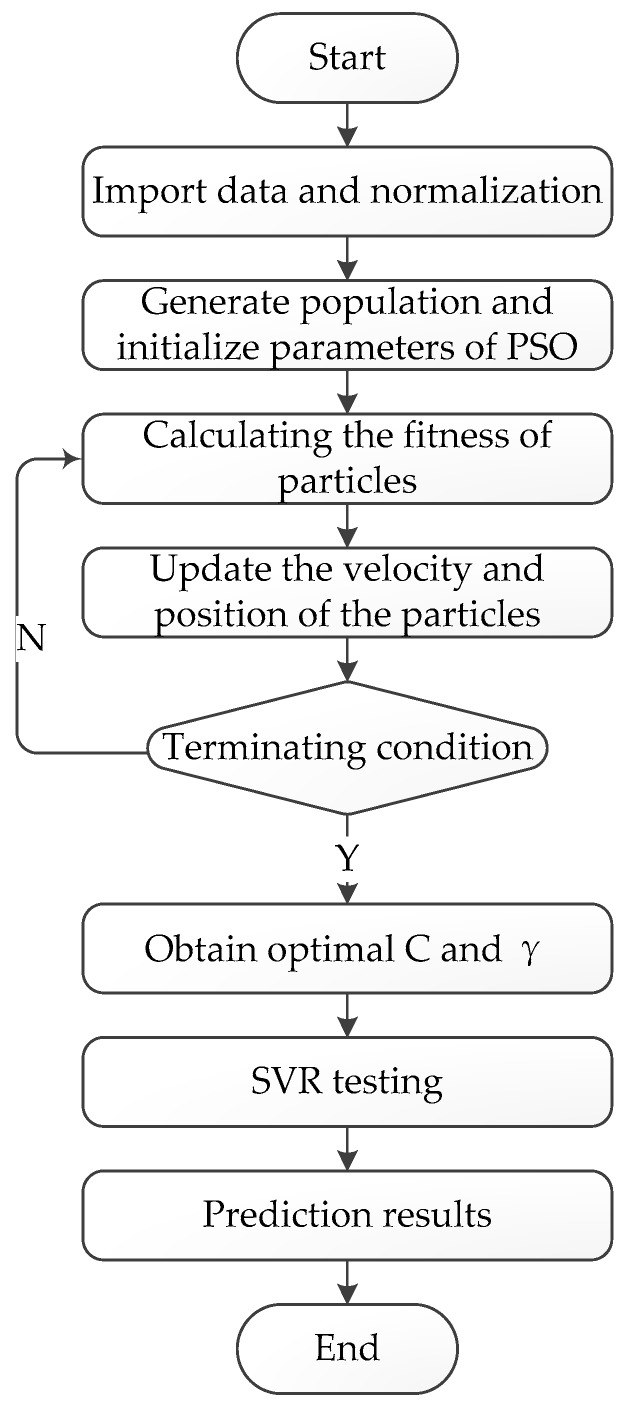
Particle swarm-optimized support vector regression (SVR) algorithm flow chart.

**Figure 3 sensors-19-03917-f003:**
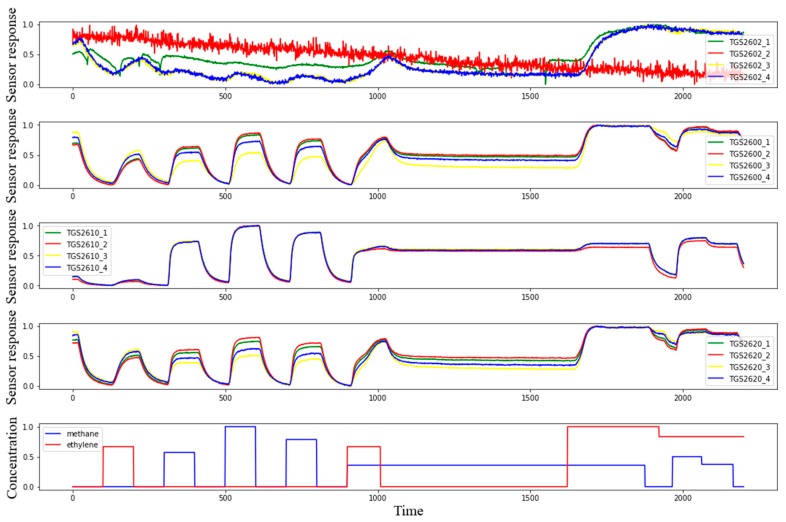
The sensor responses and gas concentrations.

**Figure 4 sensors-19-03917-f004:**
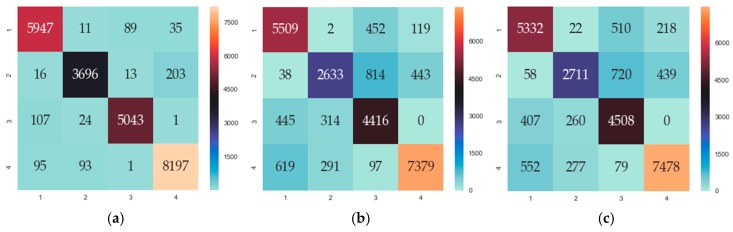
The confusion matrix of the three algorithms (1: single methane; 2: single ethylene; 3: air; 4: mixture). (**a**) The confusion matrix of the random forest (RF) classifier. (**b**) The confusion matrix of the logistic regression (LR) classifier. (**c**) The confusion matrix of the support vector machine (SVM) classifier.

**Figure 5 sensors-19-03917-f005:**
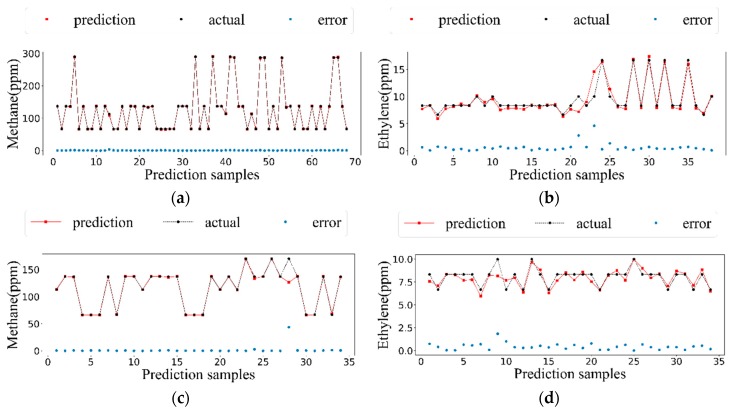
The predicted concentration along with the actual value. (**a**) The prediction and actual concentrations of single methane. (**b**) The prediction and actual concentrations of single ethylene. (**c**) The prediction and actual concentrations of methane in mixture. (**d**) The prediction and actual concentrations of ethylene in mixture.

**Figure 6 sensors-19-03917-f006:**
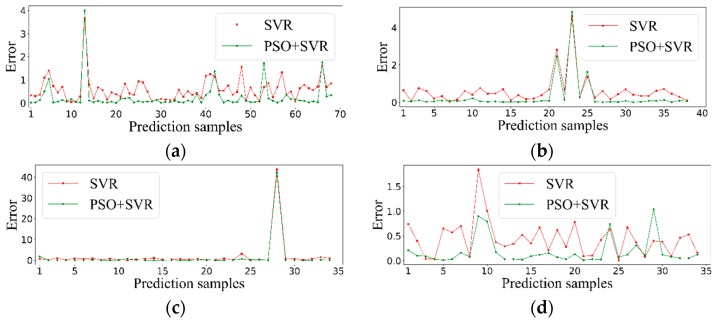
Prediction error provided by particle swarm optimization (PSO) + SVR (green) and SVR (red). (**a**) The prediction error of single methane. (**b**) The prediction error of single ethylene. (**c**) The prediction error of methane in mixture. (**d**) The prediction error of ethylene in mixture.

**Table 1 sensors-19-03917-t001:** Summary of gas mixture analysis methods.

Methodology	Application
Two-dimensional wavelet transformation feature extraction + linear-SVM classifier [26]	QAL-ID for gas mixture
PCA and partial least squares (PLS) feature extraction + SVM, RF, extreme learning machine (ELM) [7]	QAL-ID for volatile gas of the paraffin
Fuzzy adaptive resonant theory map (ARTMAP) and linear discriminant analysis (LDA) [8]	QAL-ID for gas mixture of essential oils
Two-dimensional wavelet transformation feature extraction + PLS regression [26]	QTY-ANLS for gas mixture
Least-squares support vector machine-based (LSSVM-based) nonlinear regression [24]	QTY-ANLS for gas mixture
Reservoir computing [22]	QTY-ANLS for gas mixture
Gradient boosted regression tree [27]	QTY-ANLS for emissions of LNG bus
Long Short-Term Memory(LSTM) [28]	QTY-ANLS for gas in coal mine
SVM, RF, extreme learning machine (ELM), and partial least squares regression (PLSR) [6]	QTY-ANLS for food additives in the fruit juice
Genetic algorithm + SVR [21]	QTY-ANLS for gas chromatography
Neural network [12]	QTY-ANLS for air pollutants
Empirical wavelet transformation (EWT)-multi-agent evolutionary genetic algorithm (MAEGA)-nonlinear auto regressive models (NARX) [17]	QTY-ANLS for air pollutants
SVR [16]	QTY-ANLS for PM2.5
Principal component correlation analysis (PCCA) and LSTM [29]	QTY-ANLS for natural gas
Multiple regression (MR) and SVR [25]	QTY-ANLS for methane

**Table 2 sensors-19-03917-t002:** The characteristics of the four types of sensors.

Sensors	Sensitivity (Rate of Change for R_S_)	Stability	Detection Range (ppm)
TGS2602	0.08~0.5	Long-term stability	0~10
TGS2600	0.3~0.6	Long-term stability	0~10
TGS2610	0.5~0.62	Long-term stability	500~10,000
TGS2620	0.3~0.5	Long-term stability	50~5000

**Table 3 sensors-19-03917-t003:** The eigenvalues and contribution rate of PCA.

Principal Components	Eigenvalues	Contribution Rate/%	Cumulative Contribution Rate/%
PC1	10.134	63.33	63.33
PC2	4.204	26.27	89.61
PC3	1.277	7.98	97.59
PC4	0.336	2.10	99.69
PC5	0.025	0.15	99.85
PC6	0.016	0.10	99.95
PC7	0.003	0.02	99.97
PC8	0.002	0.01	99.98
PC9	0.002	0.01	99.99
…	…	…	…
PC16	0.000	0.00	100.00

**Table 4 sensors-19-03917-t004:** Parameters and concentration estimation results of different categories.

Categories	Single Gas		Mixed Gas	
Components	Methane	Ethylene	Methane	Ethylene
*C*	22,481	13,892	27,047	8546
*γ*	2.86	0.45	0.44	0.28
*R* ^2^	0.996	0.979	0.979	0.828

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
