# Peer review of "Quantitative and Qualitative Analysis of Multicomponent Gas Using Sensor Array"

_sensors, 2019, doi:10.3390/s19183917_

Round 1
Reviewer 1 Report
Desr authors, the topic considered in your work is rather dated even if little news are introduced.Very little is left to the sensors description and their behaviour(sensitivity, stability,resolution). also the english must be improved such as: row29(who have evaluated...)row 32 (subject is missed),row 37( cancel" have"),row 41( cancel literature and use another word such as "result") and so on...Fig.3 must be improved. in fact we do not guess from it the repeatibility feature!!
Author Response
The manuscript has undergone extensive English editing by MDPI English editing service. In the introduction part, the background and application of gas detection were added, and the qualitative and quantitative analysis methods of gases were compared. In the results section, Figure 5 is improved to show the results of the quantitative analysis more clearly. Figure 3 is enhanced to visually demonstrate the response characteristics of the sensor. In the introduction to the data set, an explanation of the sensors description and their behaviour has been added.Reviewer 2 Report
This study describes quantitative and qualitative analysis of multi-components gas using sensor array.
Some inaccuracies are not avoided that I commented and presented below:
Part 1: Introduction.
For several decades, studies on application of different type of sensors for detection of gas have been conducted.
I suggest supplementing the Chapter with additional information related to other new methods and devices in research of gas detections, for example:
“Identification of volatile organic compounds and their concentrations using a novel method analysis of MOS sensors signal. Journal of Food Science 2019”,
“A novel method for generation of a fingerprint using electronic nose on the example of rapeseed spoilage. Journal of Food Science, 2019”.
“Electronic nose with polymer-composite sensors for monitoring fungal deterioration
of stored rapeseed. International Agrophysics, 2017”.
Part: Gas Mixture Analysis Method
The Method section provides the reader with enough information to repeat the experiments conducted.
Part: Qualitative Identification Method for Gas Mixture
In the subpart Principal Component Analysis.
I have a series of questions about the Principal Component Analysis (PCA) used in the work:
On the basis of which criterion was the optimal number of main components obtained in the PCA analysis determined?
How many columns and rows had a data matrix for PCA?
Is the input matrix automatically scaled?
Please put this information in this chapter.
Part: Experiments and Results
The experimental section provides the reader with enough information to repeat the experiments conducted. The most part the Results section is well structured and the obtained data were subjected to an appropriate statistical analysis (PCA).
The results were not fully discussed. A full discussion of the results obtained with other work in this field should be carried out.
The Figure 5 should be corrected for better reception and for a better comparison of results. Please correct.
Part: Conclusion
Conclusions are synthetically described and result from the conducted research.
Author Response
The manuscript has undergone extensive English editing by MDPI English editing service. There are additional information related to other new methods and devices in research of gas detections, for example:“Identification of volatile organic compounds and their concentrations using a novel method analysis of MOS sensors signal. Journal of Food Science 2019”,
“A novel method for generation of a fingerprint using electronic nose on the example of rapeseed spoilage. Journal of Food Science, 2019”.
“Electronic nose with polymer-composite sensors for monitoring fungal deterioration of stored rapeseed. International Agrophysics, 2017”.
In the introduction part, the application of gas detection and the comparison of qualitative and quantitative analysis methods are added. In the subpart Principal Component Analysis:For the questions you asked, do the following:(1)The optimal number of main components obtained in the PCA analysis determined is determined by the cumulative contribution rate of the eigenvalue for variance rCCR, which utilizes rCCR greater or equal to 99%.(2) The data matrix of PCA was 10,476 rows and 16 columns.(3) The input matrix was artificially scaled so that the mean was 0 and the variance was 1. Theses information were included in the chapter. A full discussion of the results obtained with other work in this field have been carried out. The Figure5 have been improved for a better comparison of results.Reviewer 3 Report
The article entitled "Quantitative and Qualitative Analysis of Multi-
Components Gas Using Sensor Array ", presents a theme that already sees many publications on this and this is not particularly new. The limit of this article, however, is not the data but the way in which it is written, Unfortunately it is almost impossible to understand what it is you speak and the objective of the article is not really known which is if there was not the title.
I believe that this article is not ready to be published, I suggest that it be written again.
Author Response
The manuscript has undergone extensive English editing by MDPI English editing service. In the introduction section, the application of gas detection and the comparison of qualitative and quantitative analysis methods are added. The objective of this article is proposing a novel method for gas qualitative identification and quantitative analysis.Round 2
Reviewer 3 Report
The Article titled "Quantitative and Qualitative Analysis of Multicomponent Gas Using Sensor Array", may be at my suggestion, accepted in its present form.
thank to the authors for all the corrections made, which now make the article complete in all parts.